# How to Solve the Conundrum of Heparin-Induced Thrombocytopenia during Cardiopulmonary Bypass

**DOI:** 10.3390/jcm12030786

**Published:** 2023-01-18

**Authors:** Etienne Revelly, Emmanuelle Scala, Lorenzo Rosner, Valentina Rancati, Ziyad Gunga, Matthias Kirsch, Zied Ltaief, Marco Rusca, Xavier Bechtold, Lorenzo Alberio, Carlo Marcucci

**Affiliations:** 1Department of Anesthesiology, Lausanne University Hospital (CHUV), 1011 Lausanne, Switzerland; 2Faculty of Biology and Medicine, University of Lausanne (UNIL), Rue du Bugnon 21, 1011 Lausanne, Switzerland; 3Department of Cardiac Surgery, Lausanne University Hospital (CHUV), 1011 Lausanne, Switzerland; 4Department of Intensive Care Medicine, Lausanne University Hospital (CHUV), 1011 Lausanne, Switzerland; 5Division of Hematology and Central Hematology Laboratory, Lausanne University Hospital (CHUV), 1011 Lausanne, Switzerland

**Keywords:** cardiac surgery, cardiopulmonary bypass, intraoperative management, heparin-induced thrombocytopenia syndrome, direct thrombin inhibitor, antiplatelet therapy

## Abstract

Heparin-induced thrombocytopenia (HIT) is a major issue in cardiac surgery requiring cardiopulmonary bypass (CPB). HIT represents a severe adverse drug reaction after heparin administration. It consists of immune-mediated thrombocytopenia paradoxically leading to thrombotic events. Detection of antibodies against platelets factor 4/heparin (anti-PF4/H) and aggregation of platelets in the presence of heparin in functional in vitro tests confirm the diagnosis. Patients suffering from HIT and requiring cardiac surgery are at high risk of lethal complications and present specific challenges. Four distinct phases are described in the usual HIT timeline, and the anticoagulation strategy chosen for CPB depends on the phase in which the patient is categorized. In this sense, we developed an institutional protocol covering each phase. It consisted of the use of a non-heparin anticoagulant such as bivalirudin, or the association of unfractionated heparin (UFH) with a potent antiplatelet drug such as tirofiban or cangrelor. Temporary reduction of anti-PF4 with intravenous immunoglobulins (IvIg) has recently been described as a complementary strategy. In this article, we briefly described the pathophysiology of HIT and focused on the various strategies that can be applied to safely manage CPB in these patients.

## 1. Introduction

The administration of unfractionated heparin can lead to a drop in platelet count through several mechanisms. The condition is known as heparin-induced thrombocytopenia (HIT) and is divided in two types [1]: Type 1 HIT, also named heparin-associated thrombocytopenia (HAT) is characterised by a mild drop in platelet count occurring within one to four days of heparin exposure, spontaneous recovery of platelet count without interruption of heparin treatment and is not associated with complications. Type 2 HIT represents a rare, immune-mediated, severe adverse drug reaction after heparin administration, and is the subject of this article. It is due to the development of auto-antibodies against a complex formed by platelet factor 4 (PF4) and heparin, inducing platelet activation and in vivo thrombin generation [2]. Paradoxically, clinical manifestations include potentially life-threatening thrombotic events [3]. As thrombocytopenia could have various causes, HIT diagnosis sometimes represents a true challenge. However, early recognition is fundamental, especially for patients who need surgery. The 4T score and rapid laboratory test with detection of anti-PF4/H antibodies, associated with platelet aggregation in the presence of heparin in functional assays, confirm the diagnosis. Standard treatment consists of heparin discontinuation and prompt anticoagulation with a non-heparin anticoagulant.

Nearly 50% of patients with confirmed HIT will suffer thromboembolic complications. If left untreated, HIT’s rate of thromboembolism, amputation and death is as high as 6.1% per day [4]. The most frequent complications are venous thrombosis of the large vessels of the lower limbs and pulmonary embolism [5]. HIT becomes a major issue in cardiac anaesthesia where full-dose anticoagulation is mandatory. In fact, thrombin generation and platelet aggregation in the cardiopulmonary bypass (CPB) circuit would be cataclysmic. Prevention of intraoperative thrombosis is mandatory. Thus, cardiac surgery using CPB in patients with HIT has a high risk of lethal complications. Unfortunately, strong evidence for possible management strategies during cardiopulmonary bypass is lacking, and guidelines concerning the perioperative management of these patients are vague.

In this article, we briefly described the pathophysiology of HIT, we presented evidence for the different strategies available and we focused on the rationale and practical aspects of our protocol for the intraoperative management of cardiac surgery with CPB. Pre-and postoperative anticoagulation strategies were beyond the scope of this article.

## 2. Pathophysiology of HIT

The HIT mechanism is complex and goes beyond classical platelet activation [6]. In 2022, Marchetti and colleagues reviewed new concepts in pathogenesis [7]. They describe each step of HIT pathophysiology, beginning with antigen formation. When activated, platelets secrete a large amount of platelet factor 4 (PF4), a chemokine located in their α-granules. Positively charged PF4 with negatively charged heparin form a PF4/Heparin (PF4/H) complex [8]. The HIT antigen is the neo-epitope created by conformational changes on PF4 when forming the PF4/H complex. The size and charge of the complex, depending on relative amounts of PF4 and heparin, seem to play a major part in pathogenicity. The second step is anti-PF4/H antibody synthesis. Both innate and adaptive immunity are involved. The PF4/H complex can activate thee complement, whose degradation products, C3c and C4, mediate the PF4/H complex to bind on marginal zone B-cell receptors, leading to anti-PF4/H antibodies production [9]. Circulating Immunoglobulin G class antibodies are able to link to the HIT antigen on the PF4/H complex, creating an immune complex. The Fc-part of the latter binds to the platelet FcγRIIa receptor. The FcγRIIa receptor is a transmembrane protein carrying an Fc binding site on the extracellular side and an ITAM sequence on the cytoplasmic side (Figure 1A).

Interaction between the HIT immune complex and FcγRIIa receptor leads to phosphorylation of the tyrosine residue in the ITAM sequence and further downstream intracellular signaling, mostly through sarcoma family kinases. The resulting platelet activation leads to degranulation of α-and dense granules, activation of GP IIb/IIIa and eventually aggregation.

In the specific context of HIT, it also leads to the production of procoagulant platelets and procoagulant micro-particles, increasing thrombin generation [13].

Interestingly, PF4 can easily bind not only to heparin molecules but also to other negatively charged molecules, such as nucleic acid and polysaccharides on bacteria. These findings could enlighten cases of spontaneous or autoimmune HIT in the absence of heparin administration [14].

HIT is treated by discontinuation of heparin and usually relayed by another non-heparin anticoagulant. Seven days of alternative anticoagulation classically permits platelet count recovery in 90% of patients. After a median of fifty days HIT-mediated platelet activation becomes ineffective, and after a median of eighty-five days, antibodies against PF4/H complex are no longer measurable [15]. Based on clinical signs, platelet count, the presence of HIT-antibodies and the platelet activating capacity of these antibodies, Cuker described the clinical course of HIT [16]. Antibodies are detected using immunological assays and their platelet activating capacity is tested in vitro by triggering platelet aggregation in functional assays (see below). Five sequential phases are distinguished. *Suspected HIT* is considered when patients have a clinical suspicion of HIT, but laboratory test results are pending. In *acute HIT*, the diagnosis is confirmed by laboratory tests: thrombocytopenia, detection of antibodies against the PF4/H complex and functional confirmation of platelet-activating capacity. Thrombotic risk is elevated in this phase and lasts at least until the recovery of platelet count. In *subacute HIT A*, platelet count normalizes. Both antibodies against the PF4/H complex and platelet-activating capacity are still present. *Subacute HIT B* represents the absence of platelet activation in functional tests despite the presence of antibodies. Finally, in *remote HIT*, antibodies are no longer measurable. This temporal classification is important for clinicians who are brought to manage this pathology (Table 1).

Moreover, it becomes crucial for patients expecting cardiac surgery. In 2018, the American Society of Hematology (ASH) edited evidence-based guidelines for the management of HIT [17]. In the specific setting of cardiovascular surgery, the ASH recommends that interventions should be delayed until patient has *subacute HIT B* or *remote HIT*. Intraoperative anticoagulation with heparin is suggested, with avoidance of heparin treatment before and after surgery. If surgery cannot be delayed, the ASH suggests various options for *acute HIT* and *subacute A HIT*, including intraoperative anticoagulation with bivalirudin; intraoperative heparin after preoperative and/or intraoperative plasma exchange; and intraoperative heparin combined with an antiplatelet drug. Of note, these recommendations are supported by low and very low levels of evidence. Also, the accuracy of this classification highly depends on the sensitivity and specificity of the functional tests used. Three tests are classically used to confirm the platelet activating capacity of the anti-PF4/H antibodies, crucially distinguishing the *acute* and *subacute A* phases from the *subacute B* phase. In these tests, plasma or serum containing the antibodies, obtained from the patient, is mixed with platelets obtained from healthy donors. In the aggregation test, platelet aggregation is measured by light transmission aggregometry after the addition of heparin to the mixture [18]. In the serotonin-release assay (SRA), heparin-induced platelet activation is confirmed by measuring the release of radiolabeled 14C-serotonin [19]. In the heparin-induced platelet activation (HIPA) platelet aggregation in micro-titer plate wells is detected optically [20]. Due to important variability in the reactivity of the donor platelets upon stimulation of their FcγRIIa receptor, these tests can produce false negative results [21]. It is therefore our opinion that when antibodies are detected, exposure to heparin without further measures should be avoided, even if functional tests fail to demonstrate in vitro platelet activation.

The 2020 Swiss recommendations elaborated by a panel of Swiss experts belonging to the Working Party Hemostasis (WPH) of the Swiss Society of Hematology (SGH-SSH) report the use of continuous intravenous infusion of bivalirudin in *acute HIT*, as the main “non-heparin” option in patients requiring cardiac surgery and patients in an intensive care setting [22].

It is also important to note that platelet transfusions are contraindicated during the *acute* and *subacute A* phases of HIT, unless the patient is severely bleeding, for they may increase the risk of thrombosis. This complicates the management of HIT patients during cardiac surgery and underscores the importance of trying to delay surgery at least until the platelet count has recovered on one hand and of choosing an anticoagulation protocol during surgery that provides maximum platelet protection and rapid recovery of hemostasis on the other.

## 3. Standard Heparin Protocol

Before the initiation of CPB, systemic anticoagulation is necessary. Usually, a large bolus dose of unfractionated heparin (300 to 500 IU/kg) is administered before aortic cannulation for initiation of CPB [23]. Heparin sensitivity is determined by patient-specific characteristics [24]. To prevent clot formation in the CPB circuit, adequacy of systemic anticoagulation is confirmed using the activated whole blood clotting time (ACT). ACT testing is performed approximately three minutes after heparin administration. A minimum ACT value ≥ 400 s is targeted before initiation of, and during, the entire CPB run. Although evidence defining optimal ACT is lacking, a few studies suggest that lower ACT values seem to be safe and effective [25,26] depending on the specifics of the circuits used [27].

The classical benefits of heparin use are the ability to easily titrate its anticoagulant effect and quickly reverse this effect with protamine. Other advantages include physician ease-of-use due to decades of practice, and low cost compared with alternatives.

## 4. Alternative Protocols

The standard heparin-based protocol cannot be used for patients suffering acute or subacute type HIT undergoing surgery requiring CPB. Two alternative strategies are proposed. The first is to replace heparin with a non-heparin anticoagulant. The second is the addition of a potent antiplatelet agent to the standard heparin protocol.

### 4.1. Non-Heparin Anticoagulants

Substitution of heparin with an alternative non-heparin anticoagulant in cardiac surgery has been described with several molecules, only two of which, bivalirudin and argatroban, are proposed as viable alternatives [28] (Figure 2).

Pharmacologically, these drugs act as direct thrombin inhibitors (DTI) and have relative short half-lives. In 2021, an American survey from Wanat-Hawthorne et al. found that 75% of the respondents choose bivalirudin as an alternative anticoagulant [29], following the 2018 guidelines of the Society of Thoracic Surgeons (STS), the Society of Cardiovascular Anesthesiologists (SCA) and the American Society of Extracorporeal Technology (AmSECT) [30].

#### 4.1.1. Bivalirudin

Bivalirudin is a DTI commonly used for percutaneous coronary intervention and has been studied in multiple trials with patients undergoing off-pump coronary artery bypass grafting, cardiopulmonary bypass, LVAD implantation, TAVI procedure, and pre- and postoperative ECMO support. It has a unique pharmacological profile. Its pharmacokinetic proprieties include a near-immediate onset of action after intravenous administration and a specific, non-competitive binding at two separate loci on the thrombin molecule (one at the catalytic site, the other at the level of fibrinogen binding site). Its anticoagulation effect is the consequence of the occupation of the catalytic binding site, which prevents cleavage of fibrinogen to fibrin by thrombin. Bivalirudin is able to inhibit circulation thrombin as well as fibrin-bound thrombin [31]. Its elimination half-life is about 25 min in patients with normal renal function (the shortest of the DTI). It is mainly cleared by proteolytic cleavage by its own substrate (80%). The proteolytic cleavage of bivalirudin is catalyzed by thrombin itself at its catalytic site, liberating thrombin and ending its inhibition. (Figure 2). Approximatively 20% of it is eliminated by the kidneys [32,33] and the dose needs to be adjusted in the case of renal disease.

When administered for prophylactic or therapeutic anticoagulation, monitoring of bivalirudin is performed with the activated partial thromboplastin time (aPTT). When administered for intraoperative anticoagulation, monitoring is performed with the activated clotting time (ACT). For cardiac surgery with CPB, established recommendations target a 2.5-fold prolongation of the baseline ACT or ACT ≥400 s [34].

Bivalirudin was first compared to heparin for cardiac surgery in 2004, by Merry et al., in one hundred patients undergoing off-pump coronary artery bypass grafting [35]. The authors did not find any significant differences in terms of postoperative blood loss between groups. In 2006, Smedira realized the larger EVOLUTION-OFF trial for the same intervention in one hundred and fifty-seven patients [36]. Their study concluded that bivalirudin was an effective anticoagulant, without excessive bleeding, its safety profile being similar to heparin. Following these encouraging conclusions, bivalirudin has been assessed in patients undergoing on-pump cardiac surgery. The EVOLUTION-ON trial compares in 150 patients, undergoing cardiac surgery with CPB for CABG and valve replacement, the efficacy and safety profile of bivalirudin versus the heparin/protamine standard [37]. The primary outcome was the success rate of the intervention defined by patient survival, lack of myocardial infarction, stroke, and the need for emergent revascularization. The results illustrated a comparable success rate between strategies. Of note, in the bivalirudin group, early (2 h) postoperative bleeding was significantly higher. This difference did not persist after 24 h. The conclusion of the authors was that bivalirudin is a safe and efficient drug for patients undergoing cardiac surgery with CPB. It is important to specify that these two trials did not include patients with HIT.

Then, the safety and efficacy of bivalirudin were assessed in 2007, in the CHOOSE-OFF trial, in a specific population of fifty-one patients carrying HIT antibodies, undergoing off-pump cardiac surgery [38]. The safety and efficacy profiles were comparable to the results of the study mentioned above in non-HIT patients. Simultaneously, the CHOOSE-ON trial assessed bivalirudin in 50 patients suffering HIT or with a history of HIT undergoing cardiac surgery with CBP [39]. The results and conclusion correlated with the EVOLUTION-ON trial.

It is relevant to add that, as mentioned by Koster et al. in the CHOOSE-ON Trial, the techniques of cardiac surgery and perfusion differ to the standard practice when bivalirudin is used as a substitute for heparin [39]. Indeed, as previously described, bivalirudin undergoes proteolytic cleavage by thrombin and other circulating proteases. The levels of the drug rapidly diminish in areas of stagnant blood (in the patient/circuit), or areas disconnected from systemic blood flow, leading to thrombus formation [40]. In 2022, a protocol for bivalirudin anticoagulation has been published by Sharma [41]. Modifications include PVC tubing instead of heparin-coated tubings; the addition of 50 mg of bivalirudin in the circuit; sanguineous cardioplegia reserved for short interventions; and regular flushing of shunt lines containing blood. Surgical practices are recommended to be adjusted with caution due to the particular pharmacology of bivalirudin. Morshuis et al. reported in 2013 an example of surgical adjustment [42]. They described a modified technique for implantation of a left ventricular assist device in patients suffering HIT. In classical implantation, after insertion of the inflow cannula, the ventricle is loaded, and the device is de-aired in an antegrade way. Then, the outflow graft is clamped during its anastomosis to the aorta. With this procedure, the blood in the device is disconnected from the systemic circulation. In the modified strategy, the anastomosis of the outflow graft is done first, and after unclamping, the device is de-aired retrogradely. The goal is to maximally reduce the time of disconnection of the blood in the device from the systemic circulation (60–90 s).

As mentioned by Koster et al., major bleeding due to the inability to reverse their anticoagulant effects is the principal problem with these drugs [28]. Moreover, studies revealed an inadequate anticoagulant effect, likely due to inexperience, with clotting of stagnant blood in the cardiopulmonary bypass (CPB) circuit, cardioplegia line, ventricular assist devices [43,44], or within the chest itself during cardiac surgical procedures [45].

#### 4.1.2. Argatroban

Argatroban is a small molecule that reversibly binds to the active enzyme site of thrombin and acts as a direct and highly selective inhibitor (Figure 2). It inhibits fibrin formation, thrombin-mediated platelet activation and the formation of the thrombin-thrombomodulin complex. Argatroban inhibits both circulating and cloth-bound thrombin [46]. Its onset of action is immediate with an elimination half-life of about 60 min [47]. Due to hepatic metabolism and elimination, mainly via hydroxylation and aromatization, the elimination half-life extends in the case of liver dysfunction [48]. Doses need to be reduced in cases of cytolytic disease, and even more in cholestatic disease [47]. This renders it unsuitable for use in many critically ill patients. Argatroban is classically monitored by aPTT, or Thrombin Time; the latter correlates better with argatroban plasma concentrations. Intraoperative monitoring can be performed with ACT [47].

Argatroban was successfully described in a few articles in patients who underwent off-pump coronary artery by-pass. The small amount of existing literature on the use of argatroban during cardiac surgery requiring CPB is worrisome. In 2015, Hillebrand et al. published the results of seven patients suffering from HIT who had been equipped with an LVAD under treatment of argatroban. Six patients had successful implantation, but four of them needed postoperative re-exploration due to major bleeding. One patient developed a severe thrombotic complication, as a massive intraventricular thrombus formed [44]. In 2013, a case report of Tanigawa described a fatal outcome due to failure of hemostasis in a patient with HIT, treated with argatroban, undergoing a mitral valve replacement with CPB [49]. In 2007, Martin et al. presented a case report and reviewed data concerning cardiac surgery with CPB and argatroban administration. Six cases were described, all required larger volumes of perioperative blood products and three suffered severe coagulopathy [45].

Because of the high rate of reported perioperative bleeding, we advocate against the use of argatroban for anticoagulation during cardiovascular surgery. In nonsurgical and postoperative settings, the use of argatroban appears to be reasonable [25,39].

### 4.2. Heparin Associated with Potent Antiplatelet Agents

The goal of this approach is to maintain heparin as the anticoagulant during CPB, while preventing HIT related platelet activation and aggregation. The rationale is to benefit from the advantages of the standard heparin-based protocol such as ease of use, ease of monitoring, protamine reversibility, and widespread experience. Yet the pharmacologic characteristics of the antiplatelet drug chosen need to be well understood and their potential side effects need to be balanced against the advantages.

Three classes of antiplatelet drugs are reported to have been used with success in this setting: prostacyclin receptor agonists; GP IIb/IIIa antagonists; and a P2Y_12_ receptor antagonist.

The prostacyclin (IP) receptor (Figure 1B) is a protein G_s_-bound transmembrane complex [11]. The conformational change of protein G_s_ resulting from receptor stimulation activates an intracellular signaling pathway that leads to increased activity of adenylyl cyclase (AC) and increased concentrations of cyclic adenosine monophosphate (cAMP). cAMP is a strong intracellular inhibitor of several activation and aggregation pathways such as, calcium mobilization, degranulation, and integrin receptor expression. Prostacyclin analogues are therefore potent antiplatelet drugs, capable of preventing platelet activation by most activators such as ADP, thromboxane, thrombin and HIT immune complexes.

Glycoprotein (GP) IIb/IIIa is the integrin by which platelets attach to fibrinogen and polymerized fibrin stands, as well as von Willebrand factor [50]. Its enhanced surface expression and activation is one of the final stages of platelet activation and allows platelets to aggregate. GP IIb/IIIa antagonists allow for profound inhibition of aggregation of platelets activated through all of the activation pathways, including HIT.

Finally, the P2Y_12_ receptor is one of the platelets receptors for adenosine diphosphate (ADP). It is coupled to protein Gi which undergoes conformational change upon receptor stimulation and initiates intracellular signaling pathways through phosphoinositide 3-kinase leading to dense granule secretion and integrin expression [12]. The P2Y_12_ receptor also inhibits AC, lowering the concentration of cAMP. The ADP stimulated secretion of granules containing ADP and calcium, generates a positive feedback loop in which activated platelets rapidly activate other platelets in their vicinity. P2Y_12_ receptor antagonists, specifically inhibit ADP induced platelet activation, but do not prevent activation by other activators such as thrombin, thromboxane and possibly HIT complexes [51]. Yet the overall downregulation of platelet reactivity, by interrupting the ADP dependent feedback loop and by upregulating AC and described as “platelet anesthesia”, may well suffice to prevent HIT-related complications during CPB [21].

#### 4.2.1. Prostacyclin Receptor Agonists

##### Epoprostenol

Epoprostenol, or prostacyclin (PGI2), has a very short half-life of 6 min, and acts as a potent pulmonary and systemic vasodilator and causes inhibition of platelet aggregation. It is mainly used in the setting of pulmonary hypertension and right heart failure. The literature on the use of epoprostenol in the setting of our topic is extremely sparse and the latest report dates from 2006. In 2018, Deshpande et al. reviewed the pharmacology and the use of prostacyclin analogues in the management of HIT during cardiac surgery [52]. The authors reported on one small retrospective comparative study (n = 6), a small case series (n = 3) and a case report, describing the experience with epoprostenol, totaling 10 patients. In the retrospective study, the epoprostenol infusion was started at 5 ng/kg/min and increased to 30 ng/kg/min over a period of 30 min before the administration of heparin [53]. After protamine reversal, the infusion was again gradually weaned over 30 min. No significant bleeding or thromboembolic event were reported. However, the authors mentioned systemic hypotension as a potential problem.

##### Iloprost

Iloprost is the best-studied drug of this class. It is a synthetic analogue of PGI2. Iloprost is a more stable compound and has a longer half-life of 15–30 min, compared to only six minutes for natural PGI2 [52]. It is primarily metabolized through hepatic β-oxidation, and its half-life is prolonged in patients with hepatic dysfunction and in patients with severe kidney disease requiring dialysis. It inhibits platelet aggregation and adhesion, as well as the platelet release response by increasing adenylate cyclase (AC) activity. Platelet reactivity recovers between 2 to 3 h after drug discontinuation [54].

Iloprost is not a selective IP receptor agonist and has affinity for other prostanoid receptors [55], which are present in most tissues and regulate various functions. This explains the variety of contraindications, such as asthma, COPD, pulmonary oedema, hepatic disease and pregnancy, and the variety of side effects that occur during intravenous infusion, such as flushing, headache, nausea and vomiting, back pain, jaw spasm, dyspnoea, wheezing and many more. Most of these are of no concern during general anaesthesia. However, its relaxing effect on vascular smooth muscle leads to dose-dependent vasodilation with hypotension and compensatory tachycardia. These hemodynamic effects are most notable when the dose is rapidly increased, and are mitigated by receptor desensitization when the dose is gradually increased [56]. When used for the treatment of pulmonary hypertension, the recommended starting dose is 0.5 ng/kg/min, which is slowly up-titrated over a period of months. The highest doses after one year of treatment are in the 2–2.5 ng/kg/min range [57]. Yet, at this dosing level, platelet inhibition is incomplete and likely insufficient for the safe use of heparin in the context of HIT and CPB. Krause et al. found that at an infusion rate of 3 ng/kg/min ADP-induced platelet aggregation was reduced by only 50% [54]. Moreover, they also reported that it took 135 min for platelet function to return to baseline.

In 2015, Palatianos et al. reported on a retrospective analysis of the biggest series of patients in whom iloprost was used during cardiac surgery [58]. Out of 17,000 patients, over an 11-year period, 530 were suspected of HIT and screened with ELISA and aggregation tests. One hundred and ten patients were found to have functional HIT antibodies and underwent cardiac surgery using a heparin and iloprost scheme. The target dose of iloprost was defined as the dose that reduced aggregation to less than 5% of baseline. Because of high inter-individual variability, the adequate dose needed to be determined preoperatively for every patient, using heparin-induced aggregation tests. Intraoperative infusion doses ranged from 3 to 24 ng/kg/min. At the lower dose, 95% platelet inhibition was only reached in 21 out of 110 patients. In five patients, perfusion doses had to be increased to ≥48 ng/kg/min, and the ensuing hypotension was be uncontrollable. Surgery in these patients was postponed until the antibody titres came down to allow for adequate inhibition with lower doses. Although the authors concluded that the use of iloprost is safe in cardiac surgery, they do mention that hypotension was a problem, with a mean systolic pressure drop of 37 mmHg requiring the unusually high doses of 1–4 µg/kg/min of norepinephrine.

Because of the many contraindications, and because of the complexity of the need for individual and slow-dose titration, and because of the high rate of severe vasoplegia, we do not include prostacyclin analogues as a treatment option in our institution.

#### 4.2.2. GP IIb/IIIa Antagonists

The only reported molecule of this class used to prevent HIT during cardiac surgery is tirofiban. The drug is most commonly used in patients suffering acute coronary syndrome requiring interventional cardiology [59].

Tirofiban is a reversible inhibitor of the fibrinogen receptor GP IIb/IIIa, one of the most important platelet receptors. Tirofiban inhibits fibrinogen binding to the GP IIb/IIIa receptor, and thus limits platelet adhesion and aggregation. Fibrin plays a central role by simultaneously binding to the activated GP IIb/IIIa receptors of multiple platelets leading to their cross linkage and forming the primary platelet clot, which will be eventually stabilized by cross-linked fibrin at the injury site. After intravenous injection, 65% of tirofiban is bound to plasma proteins. Fifteen minutes after a 10 µg/kg bolus injection followed by a 0.15 µg/kg/min infusion platelet aggregation is reduced by 83% [60]. Its elimination half-life is around 1.5–2 h in healthy volunteers [61], and platelet aggregation returns to baseline levels 4 to 8 h after cessation of the infusion. Tirofiban undergoes limited metabolization, is primarily excreted unchanged by the kidneys, requires dose adjustments for patients with a creatinine clearance less than 60 mL/min, and can be eliminated by hemodialysis.

Although the use of tirofiban was first proposed for this purpose over 20 years ago, the literature describing its efficacy remains very sparse. It is limited to some case reports and two larger reports [62,63,64,65]. A one-year experience with tirofiban and heparin during cardiopulmonary bypass in 47 patients with HIT type II was published in 2001 [64]. The authors administered tirofiban 10 min before the administration of heparin as a 10 µg/kg bolus, followed by a continuous infusion of 0.15 µg/kg/min. Its administration was stopped one hour before discontinuation of CPB. The patients received r-hirudin for postoperative thromboprophylaxis. The authors reported postoperative blood loss and transfusion requirements comparable to their institutional standards. None of them required surgical re-exploration because of postoperative hemorrhage. In the same year, the same group published a series of 10 patients with impaired renal function and HIT undergoing complex procedures such as redo surgery, multiple valve surgery and pulmonary artery endarterectomy, using the same protocol [65]. The preoperative creatinine clearance of the patients ranged from 29 to 44 mL/min. In spite of the high complexity of the procedures, and in spite of the fact that no dose adjustments of tirofiban were made for the reduced renal function, the authors reported minimal postoperative blood loss and transfusion requirements, no re-interventions for hemostasis and normal lengths of stay in the hospital, for all patients.

Finally, in one report, prolonged strong inhibition of aggregation using tirofiban was held responsible for massive bleeding after redo cardiac surgery in a patient with acute renal failure [63]. In this patient, on renal replacement therapy with subacute HIT, emergency redo mitral valve replacement was necessary on postoperative day 13 after combined CABG and mitral valve replacement. After protamine reversal there was refractory bleeding due to persistent platelet dysfunction requiring the administration of rFVII to promote hemostasis.

#### 4.2.3. P2Y_12_ Receptor Antagonists

Recently, the intravenous P2Y_12_ receptor antagonist cangrelor has been used in combination with heparin for HIT during CPB.

Cangrelor, a synthetic ATP analogue, is a very potent, direct, reversible P2Y_12_ receptor inhibitor. After a bolus dose of 30 μg/kg followed by a continuous infusion of 4 μg/kg/min it blocks ADP-induced platelet activation and aggregation by 99% [66]. It is characterized by a rapid onset of action (less than 2 min), and an extremely short half-life of 3 to 6 min. Within 60 min of cessation, platelet function returns to baseline [67]. Cangrelor is metabolized by intravascular dephosporylation through ecto-ADP-ases, making its elimination independent of renal or hepatic function, sex and age [68]. It is possible to rapidly and reliably monitor the antiplatelet effect of cangrelor by serial point-of-care platelet function assays throughout surgery. Its unique pharmacokinetic and pharmacodynamic properties make this drug ideal for intraoperative platelet inhibition during cardiac surgery.

Its first use during CPB in the setting of HIT has only been reported as recently as 2019, in several case reports and a case series of ten patients [69,70,71,72,73]. In all the case reports, the patients were in either the subacute B or remote HIT phases, thus implying the absence of potent HIT antibodies capable of inducing platelet activation in in vitro aggregation tests. Cangrelor was administered as a 30 μg/kg bolus immediately followed by a 4 μg/kg/min infusion. After confirmation of adequate platelet inhibition, using ADP-based point of care aggregation test, heparin was administered in the standard dose of 300–400 IU/kg. Cangrelor was stopped just prior to, or at the start of, protamine administration. In 2020, Gernhofer et al. presented a case series of ten patients who required urgent cardiac surgery (mainly pulmonary thrombendarterectomy) in a context of HIT [73]. They applied the same protocol including the loading dose of 30 μg/kg followed by the continuous infusion of 4 µg/kg/min. The infusion rate was titrated down to a rate between 2 and 4 μg/kg/min using the VerifyNow^®^ assay during CPB to optimize platelet inhibition. Ten minutes before protamine administration, cangrelor infusion was stopped.

In all the described cases, the authors mention good results and report one death in a patient suffering advanced intracardiac malignancy. All authors concluded that the use of cangrelor with heparin may be a safe and effective alternative anticoagulation strategy for patients requiring urgent cardiovascular surgery with CPB.

It is noteworthy that out of all the cases described, all but three were in the subacute or remote HIT phases. In two out of these three cases, bleeding and coagulopathy were respectively reported. The evidence of the capacity of cangrelor to prevent platelet aggregation in the presence of functional HIT antibodies remains, therefore, very weak.

We studied the effect of cangrelor on heparin-induced aggregation using classical light transmission aggregometry (LTA) in an in vitro study in 2020 [51]. Plasma from 22 patients having confirmed functional anti-PF4/H antibodies were mixed with platelet-rich plasma from healthy donors and aggregation was triggered by adding heparin. The addition of cangrelor prior to heparin reduced median aggregation by 91%. Yet, the recommended >95% inhibition [58], was only seen in 45% of the samples, whereas in 14% cangrelor did not inhibit in vitro heparin-induced aggregation at all. Because of the unpredictable inhibitor effect in individual patients, we advocate against the use of cangrelor as a standard for HIT patients undergoing cardiac surgery and recommend assessing its efficacy in vitro prior to surgery, unless a pretreatment with IVIG can be applied (see below).

### 4.3. Intravenous Immunoglobulin (IVIG)

The administration of immunoglobulins has been performed for decades as a means to treat deficiencies or to modulate autoimmune disorders. Since 1989, its capacity to reduce thrombotic complications in autoimmune HIT has been recognized, and it is increasingly used as an adjunct treatment for acute HIT [74,75]. IVIGs are obtained by pooling concentrated immunoglobulins, more specifically IgG, from thousands of human donors. The IgG in IVIG bind to all the Fcγ receptors of all cell types and the way by which IVIG prevents thrombosis in HIT is not fully understood. Several mechanisms have been proposed [76]. A detailed description of these is beyond the scope of this paper, but in short, competitive inhibition for the FcγRIIa receptor, masking of the Fcγ RIIa receptor by accumulation of IgGs and downregulation of the immune response by simultaneously activating inhibitory Fcγ receptors probably all play a part in their therapeutic action. The protective action of IVIG against HIT-related complications is reported to last up to 10 days [21].

IVIG is to be administered at the high dose of 1 g/kg given on two consecutive days as an adjunct treatment for HIT or as a preventive measure before re-exposure to heparin for cardiac surgery.

Warkentin summarized all the literature available on the use of IVIG in HIT up to 2019 (36 patients). The author advocates IVIG as adjunct therapy, especially for patients planned to have a re-exposition to heparin. Actually, many reasons to use IVIG are evoked. They play a significant role as they inhibit the FcγRIIa-mediated HIT reaction of almost all cells; bleeding risk is not increased, and the first few postoperative days are covered, as IVIG effects last around a week.

In a 2020 case report, Koster et al. presented their institutional protocol associating high-dose IVIG with cangrelor and heparin in the management of a 64-year old male, suffering subacute HIT type A and who required emergent redo cardiac surgery, sixteen days after a coronary artery bypass with mitral valve replacement [77]. The protocol consisted of IVIG 1 g/kg (administered post anesthesia induction), cangrelor at a dose of 30 µg/kg bolus followed by a 4 µg/kg/min infusion, and a heparin bolus of 400 IU/kg ten minutes after the bolus of cangrelor. Of note, the patient required multiple transfusions encompassing six red cell concentrates and 30 IU/kg of prothrombin complex concentrates. However, no thromboembolic complications were described in the postoperative period and the patient was discharged from the hospital three weeks later. The same author corroborates this thesis in a 2021 review of the current state of knowledge on HIT in cardiac surgery [21]. Even if clinical experience is still very limited, the theoretical advantages of combining the preoperative reduction of antibody load using IVIG with intraoperative “platelet anesthesia” using cangrelor are convincing. The re-exposure to heparin during surgery, and the possible postoperative release of sequestered heparin, form tissue compartments, and may lead to a surge of HIT antibodies when the protection by cangrelor has been weaned off. For this reason, a postoperative top-up dose of 0.5 g/kg of IVIG has been advocated in case of high-titer anti-PF4/heparin antibodies [21].

## 5. Conclusions

In conclusion, various strategies to manage anticoagulation during CPB in patients with HIT have been described and can be safely used. The optimal strategy for the individual patient depends on the phase of HIT, the timing of the surgery and the experience of the caregivers. It is important for all members of the care team to be familiar with the rationale and specific aspects of the various strategies, and to create an individualized care plan. Due to the infrequency of these interventions, staff in low- and medium-volume centers will struggle to gain clinical experience and therefore clear and practical guidelines must be established and strictly adhered to. In the final section of this paper, we present the rationale and practical aspects of our institutional guideline.

## 6. How We Do It in Lausanne

### 6.1. Rationale

Based on the available literature and our research and clinical experience, we have established institutional guidelines for HIT patients who require cardiac surgery with CPB. Although bivalirudin is by far the best-studied drug, our preferred strategy is preoperative treatment with IVIG followed by the combination of cangrelor and heparin for surgery. If IVIG treatment can be completed prior to surgery, the strong inhibitory effect of ADP-induced platelet activation by cangrelor should suffice to prevent complications, even without testing the latter. If the degree of urgency does not allow for preoperative treatment with IVIG, the capacity of cangrelor to inhibit heparin-induced aggregation in the individual patient must be tested in vitro prior to surgery. In the case of a negative test or lack of time to perform the test, we use tirofiban instead. Finally, if for any reason tirofiban is contraindicated, bivalirudin is used for anticoagulation during bypass. The rationale is that cardiac surgery in patients with HIT is a rare event, which is clearly illustrated by the data from Palatianos et al. [58], where only a mere 0.6% of all cardiac procedures required an alternative anticoagulation strategy during CPB for HIT. This implies that it is virtually impossible for a team of anesthetists, surgeons and perfusionists, not working in a high-volume center, to become familiar with the more complicated procedure using bivalirudin. Heparin combined with the infusion of a short-acting antiplatelet agent, allows standard practice to be maintained, limiting the risk in often complicated, urgent procedures.

### 6.2. HIT Phases and Timing

The strategy chosen depends on the phase of HIT on one hand, and on the degree of emergency of the surgery on the other (Figure 3).

In non-urgent or elective cases with acute or subacute HIT, surgery must be postponed until HIT antibodies are undetectable, at which time a standard heparin protocol can be used for anticoagulation during CPB. For pre- and post-operative anticoagulation however, non-heparin anticoagulants need to be used. In some rare cases, anti-PF4/Hep Abs persist for more than three months, or even for years after the acute HIT episode [71,78]. All patients with a history of HIT are therefore tested for the presence of antibodies, regardless of the time elapsed since the acute episode. If antibodies persist, surgery is performed as for subacute HIT patients requiring semi-urgent surgery, without further postponement.

In semi-urgent cases, efforts need to be undertaken to delay surgery in order to allow the platelet count to recover, limiting platelet transfusion and thrombotic risk. If possible, IVIG is administered in two doses of 1 g/kg over 48 h prior to surgery. The capacity of cangrelor to inhibit heparin-induced aggregation is tested by LTA as described by our group [51]. If cangrelor insufficiently inhibits aggregation, tirofiban will be used for intraoperative platelet inhibition. IVIG will likely influence the aggregation tests, therefore, the blood samples for the test must be drawn before the administration of IVIG. In case neither cangrelor nor tirofiban can be used, intraoperative anticoagulation will be managed with bivalirudin.

Finally, in urgent cases, when there is insufficient time for IVIG treatment and aggregation tests, tirofiban with heparin, or bivalirudin in case of contraindications for the former, is used without further delay.

### 6.3. Dosing, and Practical Aspects (Table 2)

#### 6.3.1. General Aspects

All intravenous and intra-arterial lines and perfusions must be heparin-free. Heparin-based flush solutions for pressure lines must be banned.

Surgical flush solutions are prepared using sodium citrate 3.8%. A bivalirudin 0.1 mg/mL solution can also be used for this purpose.

Only non-heparin coated tubings, filters, reservoirs and oxygenators are used in the extracorporeal circuits.

Sodium citrate 4% is used for anticoagulant of blood collected and processed in cell saving devices. Note: commercial or local citrate preparations may have different concentrations, which may be less effective in preventing thrombosis in this setting. It is important to verify the composition of the preparation at hand.

**Table 2 jcm-12-00786-t002:** Dosing strategies for the various drugs.

		Cangrelor	Tirofiban	Bivalirudin
Surgical flush solution		Citrate 3.8%	Citrate 3.8%	Citrate 3.8%
Cell Saver solution		Citrate 4%	Citrate 4%	Citrate 4%
Pump prime				50 mg
Before CPB	Bolus	30 µg/kg	10 µg/kg	1 mg/kg
	perfusion	4 µg/kg/min	0.15 µg/kg/min	2.5 mg/kg/h
During CPB	perfusion	4 µg/kg/min	0.15 µg/kg/min	2.5 mg/kg/h
	bolus			0.1–0.5 mg/kg
End of perfusion		15 prior to end of CPB	1 h prior to end of CPB	At start of protamine
Bolus in circuit post CPB				50 mg
Unplanned return on CPB		Bolus 30 µg/kg Perfusion 4 µg/kg/min	Bolus 5 µg/kg	Bolus 0.5 mg/kg Perfusion 2.5 mg/kg/h

#### 6.3.2. Cangrelor-Heparin Protocol


*Preparation*


50 mg of crystalline cangrelor is diluted in 5 mL of distilled water. A volume of 5 mL is withdrawn and discarded from a 250 mL Nacl 0.9% perfusion bag, and replaced by the 5 mL of cangrelor suspension, resulting in 250 mL of a 200 µg/mL dilution of cangrelor.

30 µg/kg (=0.15 mL/kg) of patient’s’ bodyweight is withdrawn from the cangrelor perfusion bag in a 20 mL syringe and put aside.

The remainder of the perfusion bag is set up in a volumetric infusion pump and connected to a dedicated lumen of the central venous catheter.


*Drug administration before and during bypass*


After sternotomy and dissection of the mediastinal structures, on surgical demand, the content of the cangrelor syringe is administered as an intravenous bolus and immediately followed by continuous infusion at a rate of 4 µg/kg/min. Because of the extremely rapid elimination of cangrelor, there can be no delay between the bolus and start of the continuous infusion and the infusion cannot be interrupted for more than 30 s.

Heparin is administered in the usual dose of 300 IU/kg, before cannulation of the aorta, but after adequate platelet inhibition is confirmed by an ADP-based point of care aggregation test (Multiplate^®^ in our center). An ACT target of 400 s is maintained during bypass.


*End of bypass*


After weaning off bypass, and at the surgeons’ discretion, protamine is administered over a 20-min period.

The cangrelor infusion is stopped at the start of the protamine infusion. The lumen of the central venous catheter is cleared of residual cangrelor by aspiration until return of blood.


*After bypass*


Platelet function will return to baseline within 60 min and can be monitored using ADP-based aggregation tests. The effect of cangrelor will not be apparent on viscoelastic tests of hemostasis such as TEG^®^ or ROTEM^®^ [79]. In case of acute HIT, the transfusion of platelet concentrate should be avoided.

In the case of unplanned returns on bypass during hemostasis and chest closure, a new bolus 30 µg/kg of cangrelor is administered and immediately followed by the continuous infusion 4 µg/kg/min, before re-heparinization.

#### 6.3.3. Tirofiban-Heparin Protocol


*Preparation*


The ready to infuse 250 mL infusion bag contains 12.5 mg of tirofiban (50 µg/mL). It is set up in a volumetric infusion pump and connected to a dedicated lumen of the central venous catheter.


*Drug administration before and during bypass:*


After sternotomy and dissection of the mediastinal structures, on surgical demand, a loading dose of 10 µg/kg is administered as a slow bolus over 3 min, immediately followed by a continuous infusion at a rate of 0.15 µg/kg/min.

10 min after the loading dose, heparin is administered in the usual dose of 300 IU/kg, before cannulation of the aorta, but after adequate platelet inhibition is confirmed by a TRAP-based point of care aggregation test (Multiplate^®^ in our center). An ACT target of 400 s is maintained during bypass.

If the platelet count exceeds 300 G/L, the bolus dose of tirofiban is increased to 15 µg/kg.

If the patient suffers renal failure requiring dialysis, only the bolus dose of tirofiban without the continuous infusion is administered.

The continuous infusion is stopped 1 h before the anticipated end of bypass. This implies that for short interventions only the bolus dose of tirofiban without the continuous infusion is administered.


*End of bypass*


After weaning off bypass, and at the surgeons’ discretion, protamine is administered over a 20-min period.


*After bypass*


The lumen of the central venous catheter is cleared of residual tirofiban by aspiration until return of blood.

Platelet function will return to baseline over a period of 6–8 h and can be monitored using TRAP-based aggregation tests. Tirofiban effects viscoelastic tests of hemostasis such as TEG^®^ or ROTEM^®^ by suppressing cloth firmness parameters [79]. In the case of acute HIT, the transfusion of platelet concentrate should be avoided.

In case of unplanned return on bypass during hemostasis and chest closure, a new bolus 5 µg/kg of tirofiban is administered before re-heparinization.

#### 6.3.4. Bivalirudin Protocol


*Preparation*


One vial of 250 mg of crystalline bivalirudin is reconstituted with 5 mL of distilled water and further diluted with 45 mL of NaCl 0.9% in a 50 mL syringe to reach a 5 mg/mL final dilution. The syringe is set up in a syringe pump.

50 mg of bivalirudin is added to the pump prime.


*Drug administration before and during bypass:*


After sternotomy and dissection of the mediastinal structures, on surgical demand, a bolus of 1 mg/kg is administered over 3 min, immediately followed by a continuous infusion at a rate of 2.5 mg/kg/h. The ACT target is 400 s. Additional boluses of 0.1 to 0.5 mg/kg are given to maintain the target ACT level. The infusion is stopped 15 min before the anticipated end of bypass.


*Additional measures during bypass*


All stagnation of blood must be avoided.

All circuit shunts containing blood must be unclamped and flushed every 15 min.

Stagnant blood in anatomic cavities (pleura, pericardium) must be aspirated into the cell saving device continuously or at least every 15 min,

Large volumes of blood (>1 L) in the venous reservoir must be avoided.

Bretschneider crystalloid cardioplegia (Custodiol^®^) is to be preferred. If blood cardioplegia is used, a dose must be administered every 15 min.

If a hemofilter is integrated in the circuit, blood must be allowed to freely flow through the filter when filtration is not needed, by clamping the effluent line of the filtrate.


*End of bypass*


After weaning off bypass, and at the surgeons’ discretion, protamine is administered over a 20-min period.


*After bypass*


The lumen of the central venous catheter is cleared of residual bivalirudin by aspiration until return of blood.

50 mg of bivalirudin is added to the bypass circuit and the blood volume is circulated in closed circuit with all the shunts open as long as the circuit remains in standby. In the case of an unplanned return on bypass during hemostasis and chest closure, a new intravenous bolus 0.5 mg/kg of bivalirudin is administered and the continuous infusion is restarted at the rate of 2.5 mg/kg/h.

## Figures and Tables

**Figure 1 jcm-12-00786-f001:**
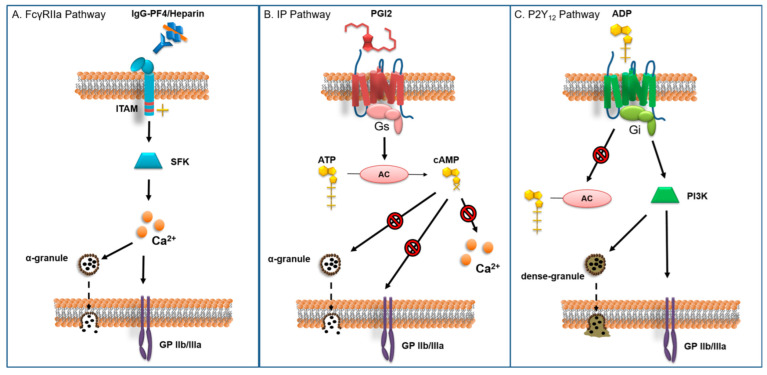
Platelet activation pathways: (**A**) FcγRIIa receptor pathway: The Fc segment of the IgG-PF4/Heparin complex binds to the extracellular segment of the FcγRIIa receptor. This leads to phosphorylation of the intracellular ITAM segment and the activation of intracellular signaling cascades through several SFKs. These trigger the liberation of intracellular Ca^2+^ stores, which in turn lead to degranulation of α-granules and activation of GP IIb/IIIa (α2β3 integrin) [10]; (**B**) IP receptor pathway: the IP receptor is bound to protein Gs which activates adenylyl cyclase upon stimulation of the receptor by PGI2. Adenylyl cyclase catalyzes the transformation of ATP into cAMP. The latter is a strong inhibitor of various intracellular activation pathways that promote Ca^2+^ release, degranulation and GP IIb/IIIa enhanced expression/activation. As such, PGI2 inhibits platelet activation by different activators such as ADP, thrombin, collagen, IgG and TXA [11]; and (**C**) P2Y_12_ receptor pathway: The P2Y_12_ receptor is coupled to the Gi protein. Upon stimulation by ADP, Gi signaling initiates an intracellular cascade through PI3K that leads to degranulation and GP IIb/IIIa expression, on one hand. On the other hand P2Y_12_ receptor activation suppresses adenylyl cyclase activity, reducing the concentration of inhibitory cAMP [12]. ITAM: immunoreceptor tyrosine-based activation motif. SFK: sarcoma family kinases. PI3K: phosphoinositide 3-kinase. PGI2: prostacyclin. TXA. Thromboxane.

**Figure 2 jcm-12-00786-f002:**
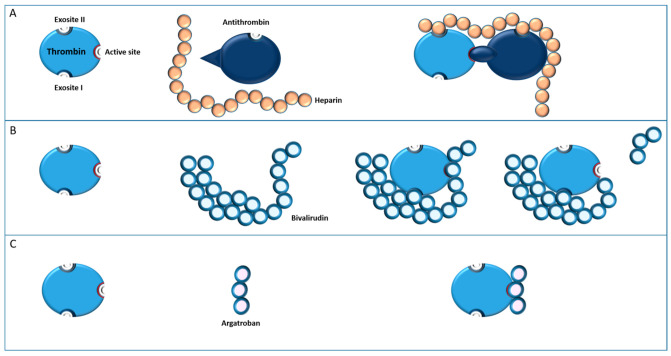
Various thrombin inhibitors with description of main sites of interest: (**A**) indirect inhibition of thrombin by heparin through the intermediate binding to cofactor antithrombin. Of note, binding of antithrombin to thrombin’s active site after heparin increases its affinity, and binding of heparin to thrombin’s exosite II: (**B**) direct inhibition of thrombin by bivalirudin with binding to both active site and exosite I. Binding leads to conformation change and proteolytic cleavage; and (**C**) direct inhibition of thrombin by small peptide argatroban with a unique bound on active site. Exosite I: fibrinogen binding site. Exosite II: heparin binding site.

**Figure 3 jcm-12-00786-f003:**
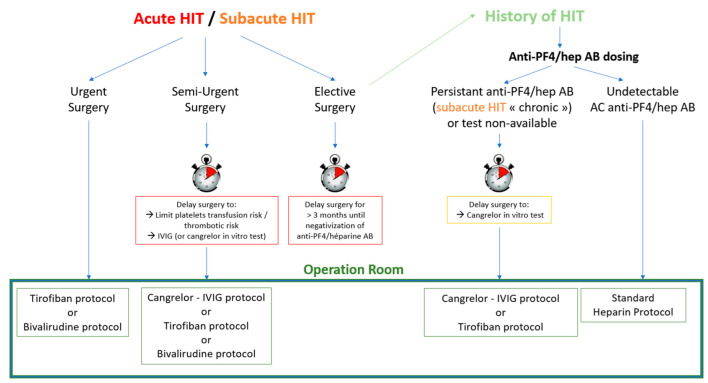
Institutional protocol describing procedure and protocol depending on HIT phase and surgical timing.

**Table 1 jcm-12-00786-t001:** Perioperative strategies depending on HIT clinical course.

HIT Phase	Postpone Surgery, If Possible	Intraoperative Anticoagulation Strategy	Pre-/Post-Operative Anticoagulation Strategy
*Acute*	Yes	Alternative protocol	Non-heparin molecule
*Subacute A*	Yes	Alternative protocol	Non-heparin molecule
*Subacute B*	No	Alternative protocol	Non-heparin molecule
*Remote*	No	Standard heparin	Non-heparin molecule

## Data Availability

Not applicable.

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
