# Peer review of "How to Solve the Conundrum of Heparin-Induced Thrombocytopenia during Cardiopulmonary Bypass"

_jcm, 2023, doi:10.3390/jcm12030786_

Round 1

Reviewer 1 Report

Overall, this is an informative, well-written review on HIT during cardiopulmonary bypass. The authors make a systematic contribution to the research literature in this area of investigation. The strategy of their own protocol is a welcome addition. Overall, this is a high quality manuscript, well done !

Minor comments: 

-       Type I HIT, also known as heparin-associated thrombocytopenia (HAT), is a non-immune mediated reaction. To be complete, I’d mention this in the introduction.

-       110 : How would you define ‘platelet activating capacity’ ? 

-       Page 5 figure 2: To be clear, I’d recommend to put the name ‘thrombine’ in the figure. For a good understanding I’d suggest to define exosite I and II as fibrinogen binding site and heparine binding site.

-       289: The prostacyclin (IP) receptor: please define IP

-       317: epoprostenol: I’d also add the half life of the product.

-       374: I’d suggest to mention the real contraindications for the product in cardiac surgery.

-       435: In many centers, the aggregation tests are not available. Could the ROTEM be of some use to detect the efficacy of the product ?

-       Considering the use of these products in HIT patients, what about the safety of platelet transfusions ? Can you safely administer platelets after CPB ? Are some products safer when you need to administer platelet transfusions ? I’d suggest to comment on this! This might be already a concern when starting a case in a patient with preoperatively a low platelet count.

-       In many centers, citrate 2,2% is being used for cell saver solutions. This might be important. I personally had a case (HIT patient) where we’ve used tirofiban and heparin but due to the low citrate solution, cloths were present. It would be good to emphasize this.

Reviewer 2 Report

Thank you very much for the opportunity to read this article. The work is very interesting, mainly presenting the center's approach to the surgical and perioperative management of patients diagnosed with HIT. I believe that the very detailed description of the treatment of patients with HIT presented in work can be beneficial, even instructional, for teams dealing with the described problem in their clinical practice, and it is undoubtedly a great starting point for discussions with groups that have their own experience and developed protocols perioperative management of patients with HIT.
As for my comments, they stem mainly from clinical curiosity and are in no way a critique of the work, which I find very interesting and well-written. However, I want to raise the following issues:
1. The authors mention in the content that thromboembolic complications in the group of patients with HIT reach 50% of patients (line 40) - I understand that this observation is consistent with the results presented in citation no. 4.
However, I would like to ask whether this percentage is so high in the practice of your center. Based on my experience, it seems that HIT occurs relatively often in the center where I work, but thromboembolic complications in this group of patients are somewhat less frequent.
2. The authors mention the use of GPIIb/IIIa receptor inhibitors and, more specifically, tirofiban. I want to ask if the authors have any experience with eptifibatide. My question stems from the fact that eptifibatide is used extensively at my facility, and access to tirofiban is very limited.
3. One of the treatment regimens described by the authors assumes the administration of an IVIG infusion and cangrelor with heparin to the Patient. I want to ask whether the authors see the possibility of modifying this regimen with the use of clopidogrel, especially in patients with HIT undergoing cardiac surgery in whom there are indications for the use of chronic antiplatelet therapy.
4. The first word on line 224 is probably 'assessed.'
